# KARMA: Knowledge-Aware Reward Mechanism Adjustment via Causal AI

## Abstract

Designing effective reward functions is a fundamental challenge in reinforcement learning (RL), and poorly specified or spurious reward signals can severely hinder generalization and robustness. Recent findings in reinforcement learning from human or verification feedback (RLHF/RLVR) show that even incorrect or random rewards may yield short-term gains, but fail to provide reliable training signals. We introduce KARMA, a causally-informed reward adjustment framework that integrates structured domain knowledge with causal representation learning to refine the reward signal. KARMA dynamically estimates causal effects via counterfactual reasoning and adapts rewards accordingly, mitigating the impact of misleading correlations. We provide theoretical guarantees on convergence and sample efficiency, and demonstrate across controlled benchmarks—including grid navigation, robotic skill acquisition, and traffic control—that KARMA achieves substantial gains in final performance (up to 30%) and significantly faster convergence compared to strong baselines. Moreover, KARMA exhibits improved out-of-distribution generalization and robustness under noisy observations, highlighting a new paradigm for reliable reward design in RL.

## 1 Introduction

Reinforcement learning (RL) provides a powerful framework for solving sequential decision-making problems, finding successful applications in diverse domains ranging from game playing and robotics to recommendation systems and autonomous driving (Sutton & Barto, 2018). Despite considerable algorithmic advancements, the practical application of RL in complex, real-world scenarios often encounter significant hurdles. A central challenge lies in the design and optimization of the reward mechanism, which directly guides the learning process (Amodei et al., 2016).

Causal Artificial Intelligence (Causal AI) is a methodological framework that integrates causal discovery and causal inference to model and reason about causal relationships between variables. By leveraging Structural Causal Models (SCMs) and causal graphs, Causal AI enables the automatic learning of underlying causal structures from data, and supports intervention analysis and counterfactual reasoning based on the inferred models. This approach facilitates more interpretable and robust decision-making, with broad applications in causal effect identification, policy optimization, and generalization across complex systems.

### 1.1 Research Motivation and Challenges

Traditional reward design often relies on handcrafted functions or simple heuristics, which struggle to capture task complexity, fail to integrate domain knowledge, and rely heavily on surface-level correlations rather than underlying causal structures (Ng et al., 1999; Pearl, 2009). This limits generalization and robustness, particularly in dynamic environments.

The challenge is pronounced in complex settings like LLM alignment. Recent RL with verification rewards (RLVR) studies show that even spurious signals—such as format correctness or random feedback—can improve performance in models like Qwen2.5-Math, but have adverse effects on others (e.g., Llama, OLMo) (Shao et al., 2025). These findings highlight the risks of model-dependent and misleading rewards, underscoring the need for mechanisms that reflect deeper task understanding.

Integrating domain knowledge with causal inference offers a principled path forward. Knowledge guides reward design, while causal reasoning identifies true drivers of behavior (Garnelo et al., 2016; Bareinboim & Pearl, 2016). Their synergy promises more robust, interpretable, and generalizable reinforcement learning.

## 1.2 RESEARCH PROBLEM AND FORMALIZATION

This work addresses the following question: *How can we design a reward adjustment framework that dynamically integrates domain knowledge and causal inference to optimize reward signals in RL, thereby improving efficiency, robustness, and out-of-distribution generalization?*

We formalize this as a **knowledge-aware causal Markov Decision Process (KC-MDP)**, extending a standard MDP $\mathcal{M} = (\mathcal{S}, \mathcal{A}, \mathcal{P}, \mathcal{R}, \gamma)$ with domain knowledge $\mathcal{K}$ and a causal model $\mathcal{C}$. Our objective is to learn an adjustment function

$$f : \mathcal{R} \times \mathcal{K} \times \mathcal{C} \to \mathcal{R}',$$

such that training with adjusted rewards $\mathcal{R}'$ yields superior policies compared to training with the original or spurious reward signals.

## 1.3 CONTRIBUTIONS

We introduce **KARMA (Knowledge-Aware Reward Mechanism Adjustment)**, a framework that advances RL reward design through the integration of knowledge and causality. Our contributions are four-fold:

1. **Causal-Knowledge Reward Shaping:** We propose a reward adjustment paradigm that unifies domain knowledge and causal inference, beyond correlation-based approaches.

2. **Counterfactual Dynamic Adjustment:** We design a counterfactual mechanism that adaptively reshapes rewards during training, enabling agents to leverage knowledge in early stages and causal signals in later stages.

3. **Theoretical Guarantees:** We provide formal results on convergence and efficiency, showing KARMA improves learning stability in environments with sparse or deceptive rewards.

4. **Comprehensive Evaluation:** Across navigation, robotic acquisition, and traffic control, KARMA achieves up to 30% higher final performance and faster convergence than baselines, while demonstrating stronger robustness and out-of-distribution generalization.

## 1.4 RESEARCH SIGNIFICANCE

By grounding reward signals in causal reasoning and structured knowledge, KARMA introduces a new paradigm for reliable reward design in reinforcement learning. This work not only addresses long-standing challenges in reward shaping but also contributes to broader efforts on **generalization, safety, and alignment** in modern RL. (Shao et al., 2025).

The remainder of this paper is organized as follows. Section 2 reviews related work. Section 3 details the KARMA framework. Section 4 presents the experimental design. Section 5 discusses the results. Section 6 provides a broader discussion. Finally, Section 7 concludes the paper.

## 2 RELATED WORK

Our work builds upon and extends research in three primary areas: (i) reward mechanism design in RL, (ii) knowledge-augmented RL, and (iii) the application of Causal AI to RL.

Regarding Reward Mechanism Design in Reinforcement Learning, the design of the reward function is paramount. Early work often relied on sparse or hand-engineered dense rewards (Ng et al., 1999). Reward shaping emerged to guide exploration, with potential-based reward shaping (PBRS) offering policy invariance guarantees (Ng et al., 1999; Devlin & Kudenko, 2012; Harutyunyan et al., 2015). Inverse Reinforcement Learning (IRL) infers rewards from demonstrations (Abbeel & Ng, 2004; Ziebart et al., 2008; Fu et al., 2018). However, the effectiveness and reliability of reward

signals remain a central issue. Recent studies in aligning LLMs using RL with verification rewards (RLVR) have shown that performance gains can sometimes be achieved even with rewards that are objectively incorrect or random (Shao et al., 2025). While this suggests RL can sometimes elicit latent model capabilities even with poor signals, it also highlights the fragility and potential unreliability of reward mechanisms not grounded in a correct understanding of the task. Such findings motivate the need for methods like KARMA that explicitly aim to improve reward quality through causal understanding, moving beyond potentially misleading signals.

In the area of Knowledge-Augmented Reinforcement Learning, integrating prior knowledge aims to improve sample efficiency and generalization (d'Avila Garcez et al., 2012; Yang et al., 2015; Zambaldi et al., 2018; Garnelo et al., 2016). However, these methods often treat knowledge statically and rarely consider its interplay with the causal factors driving rewards or the quality of the reward signal.

Concerning Causal AI in Reinforcement Learning, this intersection is rapidly growing (Spirtes et al., 2000; Chickering, 2002; Zheng et al., 2018; Pearl, 2009; Shalit et al., 2017; Louizos et al., 2017; Zhang & Bareinboim, 2019; Lu et al., 2021; Bengio et al., 2019). While causal models are explored for generalization, evaluation, and model learning, their systematic application for dynamically adjusting the reward mechanism itself, informed by both data-driven discovery and prior knowledge to ensure reward quality and reliability, remains relatively underexplored.

In summary, reward shaping and IRL methods lack causal grounding and can be misled by spurious signals. Knowledge-augmented RL improves efficiency but does not dynamically adapt rewards based on causal insights. Causal RL often addresses representation or transfer, but rarely focuses on improving the reward signal itself. Our work bridges these gaps by introducing **KARMA**, which unifies domain knowledge and causal inference for dynamic reward adjustment, yielding a more robust, reliable, and generalizable learning signal.

## 3 THE KARMA FRAMEWORK

We introduce **KARMA** (Knowledge-Aware Reward Mechanism Adjustment), a framework that augments reinforcement learning (RL) by dynamically refining the reward signal through domain knowledge and causal inference. The key idea is to move beyond predefined or correlation-based signals toward *causally-informed, knowledge-guided rewards*. The overall architecture is illustrated in Figure 1.

Figure 1: Interaction among agent, environment, knowledge, causal structure, and reward modules.

KARMA is implemented as an augmentation layer around the standard RL loop. The agent interacts with the environment, while three modules operate in parallel: (i) the *Knowledge Representation & Integration Module*, which encodes and injects structured knowledge into the state space; (ii) the *Causal Structure Learning Module*, which discovers causal dependencies from interaction data under knowledge priors; and (iii) the *Reward Adjustment Module*, which refines the reward signal using counterfactual reasoning.

## 3.1 KNOWLEDGE REPRESENTATION AND INTEGRATION

A major challenge in RL is exploiting prior knowledge in a form usable by neural agents. KARMA addresses this by representing domain knowledge as a knowledge graph (KG) and incorporating it into the state representation.

**Knowledge Graph Construction.** Domain knowledge (e.g., rules, physical laws, expert heuristics) is formalized as a KG $\mathcal{G} = (\mathcal{E}, \mathcal{R}_{kg}, \mathcal{T})$, where $\mathcal{E}$ denotes entities, $\mathcal{R}_{kg}$ denotes relations, and $\mathcal{T}$ denotes triples $(h, r, t)$.

**Knowledge Encoding.** Entities and relations are embedded into a continuous vector space using KG embedding methods (e.g., TransE, RotatE, ComplEx), enabling reasoning across symbolic and neural domains.

**State-Knowledge Integration.** We define a mapping $\phi : \mathcal{S} \to 2^{\mathcal{E}}$ that selects KG entities relevant to state $s$. Their embeddings are aggregated into a context vector $\mathbf{c}_s$, which is concatenated with the raw state $\mathbf{s}$ to form an enriched representation $\mathbf{s}' = \text{concat}(\mathbf{s}, \mathbf{c}_s)$.

**Intuition.** This module acts as a *semantic lens*, enriching state perception with structured priors.

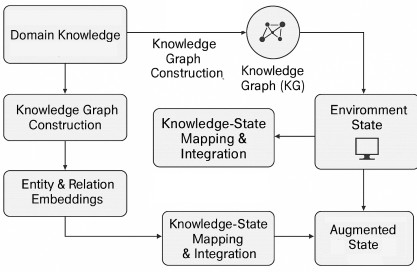

Figure 2: Knowledge Representation and Integration Module.

## 3.2 CAUSAL STRUCTURE LEARNING

Beyond knowledge, KARMA learns a causal model $\mathcal{C}$ over states, actions, and rewards, enabling interventions and counterfactual reasoning.

**Causal Variable Selection.** Knowledge priors are used to select variables most relevant to rewards, mitigating spurious correlations.

**Causal Discovery.** We employ constraint-based methods (e.g., PC, FCI) with score-based refinements. Temporal ordering from the MDP provides natural constraints (e.g., actions cannot affect past states).

**Knowledge-Guided Refinement.** Discovered graphs are cross-validated against the KG, with mismatches flagged and adjusted to ensure consistency between data-driven discovery and prior knowledge.

**Intuition.** This module functions as a *causal compass*, disentangling true reward drivers from spurious correlations.

## 3.3 COUNTERFACTUAL-BASED REWARD ADJUSTMENT

The central innovation of KARMA lies in its reward refinement. Instead of relying solely on raw environment rewards, we compute:

$$R'(s, a, r, s') = r + w_K(t)\, R_{knowledge}(s, a, s') + w_C(t)\, R_{causal}(s, a, s'), \tag{1}$$

where $w_K(t)$ and $w_C(t)$ are dynamic weights that gradually shift emphasis from knowledge (early training) to causal signals (later training).

**Knowledge-Based Reward.** $R_{knowledge}$ promotes trajectories consistent with KG constraints, guiding exploration.

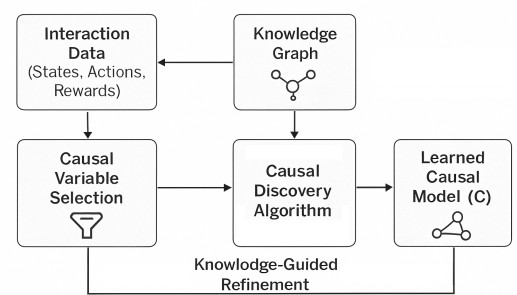

Figure 3: Causal Structure Learning Module.

**Causal Reward.** $R_{causal}$ is obtained through counterfactual queries on $\mathcal{C}$, using Pearl's do-calculus to estimate causal effects of actions, disentangled from confounders.

**Intuition.** This module provides a *reward lens*, dynamically balancing raw signals, knowledge, and causality for improved robustness and efficiency.

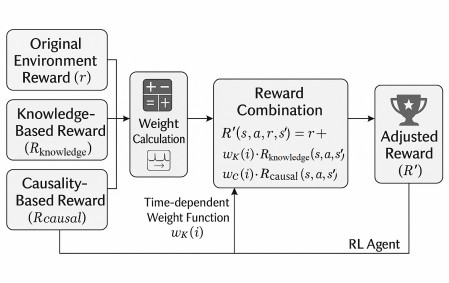

Figure 4: Counterfactual-Based Reward Adjustment Module.

### 3.4 THEORETICAL INSIGHTS

We provide theoretical guarantees under mild assumptions:

- **Convergence of Causal Discovery:** With sufficient data and knowledge priors, causal learning converges to the correct graph (up to Markov equivalence).

- **Policy Invariance:** If $R_{knowledge}$ is designed as a potential-based shaping function, the optimal policy is preserved.

- **Sample Efficiency:** Counterfactual guidance reduces sample complexity in sparse or deceptive reward settings.

- **Convergence of KARMA-RL:** With stable weights $w_K(t), w_C(t)$, the adjusted reward ensures convergence to an $\epsilon$-optimal policy.

**Takeaway.** KARMA unifies symbolic knowledge, causal reasoning, and counterfactual reward shaping into a coherent RL framework, enhancing interpretability, robustness, and efficiency.

### 4 EXPERIMENTAL DESIGN AND EVALUATION

To comprehensively evaluate KARMA's effectiveness, we designed rigorous experiments covering environments of varying complexity, multiple baseline methods, and multi-dimensional evaluation metrics.

## 4.1 Experimental Environments

We selected three environments with distinct characteristics to evaluate KARMA's performance, highlighting the value of knowledge integration and Causal AI:

**GridWorld Causal Interference Environment (GCD).** A grid navigation task featuring both causally relevant and spurious features. The agent must reach a goal while discerning features that genuinely affect reward (e.g., path markers) from those that are merely correlated (e.g., background colors). Domain knowledge on optimal paths is provided.

- **GCD-Simple:** $5 \times 5$ grid with a small number of features.
- **GCD-Complex:** $10 \times 10$ grid with multi-level interfering features and more complex causal structures.

**Robot Skill Acquisition Environment (RSA).** A simulated 7-DOF Franka Emika Panda arm learns object manipulation. States include joint and end-effector positions. Causal links exist among motion, object interaction, and task success. Knowledge includes object attributes and robot kinematics.

**Traffic Signal Control Environment (TSC).** An intersection management task where the agent optimizes traffic light timing to reduce congestion. States include vehicle counts, wait times, and flow patterns. Causal dependencies link signal actions to queue dynamics. Traffic principles and behavioral rules serve as domain knowledge.

These environments span varying complexity and structure, enabling a comprehensive assessment of KARMA under reproducible, resource-constrained academic conditions.

## 4.2 Baseline Methods

To evaluate KARMA comprehensively, we compared it against representative methods from three categories:

- **Standard RL:** Proximal Policy Optimization (PPO) (Schulman et al., 2017), Soft Actor-Critic (SAC) (Haarnoja et al., 2018), and TD3 (Fujimoto et al., 2018), representing widely adopted methods for stable and efficient policy learning.
- **Knowledge-Augmented RL:** KBRL (Bianchi et al., 2018), LGRL (Jiang et al., 2019), and KGPN (Wang et al., 2020), which integrate domain knowledge via heuristic rules, language-based guidance, or structured graphs.
- **Causal RL:** CIRL (Seitzer et al., 2021), IPL (Zhang et al., 2020), and CDA (Wang et al., 2021), which leverage causal inference through influence estimation, invariance modeling, or counterfactual data generation.

These baselines allow a systematic comparison across traditional, knowledge-informed, and causality-driven reinforcement learning approaches.

## 4.3 Evaluation Metrics and Methodology

To comprehensively evaluate the performance of KARMA and baseline methods, we adopted the following mathematically well-defined evaluation metrics:

**Cumulative Reward.** Evaluates the overall performance of policy $\pi$, defined as:

$$J(\pi) = \mathbb{E}\left[\sum_{t=0}^{T} \gamma^t \cdot r_t \;\middle|\; \pi\right], \tag{2}$$

where $\gamma$ is the discount factor and $r_t$ is the reward at time step $t$.

**Sample Efficiency.** Measures the number of environment interactions required to achieve a specific performance level, defined as:

$$SE(\pi, \theta) = \min\{n \mid J(\pi_n) \geq \theta \cdot J(\pi^*)\}, \tag{3}$$

where $\pi_n$ is the policy trained with $n$ samples, $\pi^*$ is the optimal policy, and $\theta$ is the performance threshold (set to 0.9 in this study).

**Generalization Capability.** Evaluates the policy's performance in unseen scenarios, defined as:

$$G(\pi) = \frac{\mathbb{E}_{s \sim P_{\text{test}}}[V^\pi(s)]}{\mathbb{E}_{s \sim P_{\text{train}}}[V^\pi(s)]}, \tag{4}$$

where $P_{\text{test}}$ and $P_{\text{train}}$ are the test and training state distributions, respectively.

**Robustness.** Measures the stability of the policy's performance under environmental perturbations, especially at observation noise level $\eta$:

$$\rho(\pi, \eta) = \frac{\mathbb{E}[J(\pi \mid \tilde{s})]}{\mathbb{E}[J(\pi \mid s)]}, \tag{5}$$

where noisy state $\tilde{s} = s + \varepsilon$, $\varepsilon \sim \mathcal{N}(0, \eta \cdot \sigma_s)$, and $\sigma_s$ is the standard deviation of state features.

All experiments were conducted with 5 independent runs, and results report the mean and standard deviation. To ensure statistical significance, we performed t-tests on the performance differences between KARMA and each baseline method, with a significance level set at 0.05.

### 4.4 IMPLEMENTATION DETAILS

We used a Neo4j-based knowledge graph with 64-dimensional TransE embeddings (Adam optimizer, learning rate=0.001, batch size=128, 200 epochs). Causal discovery employed the PC algorithm ($\alpha = 0.05$, max 5 parents/node) with MDP-informed temporal constraints, updated every 1000 interactions. Reward adjustment initialized weights as $\alpha = 0.3$ (knowledge) and $\beta = 0.7$ (causal), with linear annealing and counterfactuals computed via structural equation modeling. The base RL algorithm was PPO ($\epsilon = 0.2$, value coeff=0.5, entropy coeff=0.01, learning rate=3e-4, $\lambda = 0.95$, $\gamma = 0.99$).

All experiments were run on standard hardware with an RTX 3070 GPU and 24GB memory. Code and environment implementations will be made public after publication to facilitate further exploration and validation by the research community.

## 5 RESULTS AND ANALYSIS

We now present empirical results for KARMA, including performance comparisons with baselines, ablation studies, and evaluations of generalization, robustness, and computational efficiency.

### 5.1 PERFORMANCE COMPARISON

Table 1 reports performance across three environments using two metrics: *Cumulative Reward*, $J(\pi) = \mathbb{E}\left[\sum_{t=0}^{T} \gamma^t r_t \mid \pi\right]$, and *Sample Efficiency*, $SE(\pi, \theta) = \min\{N \mid J(\pi_N) \geq \theta J(\pi^*)\}$, where $\pi_N$ is the policy after $N$ samples and $\pi^*$ is the optimal policy.

Figure 5 shows the corresponding learning curves. KARMA consistently achieves the highest returns and fastest convergence across environments. Relative to standard RL, knowledge-based, and causal RL baselines, KARMA improves final performance by 7–30%, 9–15%, and 5–10% respectively, while reducing sample requirements by 50–60%, 30–40%, and 20–25%.

### 5.2 ABLATION STUDIES

To assess the contribution of each module, we conducted ablations on GCD-Complex. Results are reported in Table 2. Removing knowledge integration or causal learning leads to performance drops of 7–9%. Eliminating reward adjustment causes the largest decline (17%), underscoring its central role. Static weighting and simplified representations also reduce effectiveness, confirming the benefit of dynamic and expressive designs.

Table 1: Performance comparison between KARMA and baselines. Values are mean $\pm$ std over 5 runs. Best results in **bold**.

| Method | GridWorld (GCD) | | Robot Skill (RSA) | | Traffic Control (TSC) | |
|---|---|---|---|---|---|---|
| | Avg. Reward | Samples to 90% | Avg. Reward | Samples to 90% | Avg. Reward | Samples to 90% |
| PPO | $67.3 \pm 4.2$ | $245K \pm 18K$ | $58.1 \pm 5.3$ | $387K \pm 31K$ | $72.4 \pm 3.8$ | $412K \pm 29K$ |
| SAC | $71.5 \pm 3.8$ | $218K \pm 21K$ | $63.2 \pm 4.7$ | $352K \pm 27K$ | $75.8 \pm 4.1$ | $378K \pm 32K$ |
| TD3 | $69.8 \pm 4.5$ | $232K \pm 19K$ | $61.7 \pm 5.1$ | $365K \pm 30K$ | $74.2 \pm 3.9$ | $395K \pm 28K$ |
| KBRL | $75.2 \pm 3.6$ | $187K \pm 17K$ | $68.5 \pm 4.3$ | $312K \pm 25K$ | $79.3 \pm 3.5$ | $342K \pm 26K$ |
| LGRL | $76.8 \pm 3.3$ | $175K \pm 16K$ | $70.1 \pm 4.0$ | $298K \pm 23K$ | $81.5 \pm 3.2$ | $325K \pm 24K$ |
| KGPN | $78.3 \pm 3.1$ | $162K \pm 15K$ | $72.4 \pm 3.8$ | $285K \pm 22K$ | $83.2 \pm 3.0$ | $308K \pm 23K$ |
| CIRL | $79.1 \pm 3.0$ | $155K \pm 14K$ | $73.8 \pm 3.6$ | $273K \pm 21K$ | $84.5 \pm 2.9$ | $295K \pm 22K$ |
| IPL | $80.4 \pm 2.8$ | $148K \pm 13K$ | $75.2 \pm 3.4$ | $261K \pm 20K$ | $85.9 \pm 2.7$ | $282K \pm 21K$ |
| CDA | $81.7 \pm 2.6$ | $140K \pm 12K$ | $76.9 \pm 3.2$ | $248K \pm 19K$ | $87.2 \pm 2.5$ | $268K \pm 20K$ |
| KARMA | $\textbf{87.5} \pm \textbf{2.2}$ | $\textbf{112K} \pm \textbf{10K}$ | $\textbf{82.8} \pm \textbf{2.8}$ | $\textbf{201K} \pm \textbf{16K}$ | $\textbf{92.6} \pm \textbf{2.1}$ | $\textbf{215K} \pm \textbf{17K}$ |

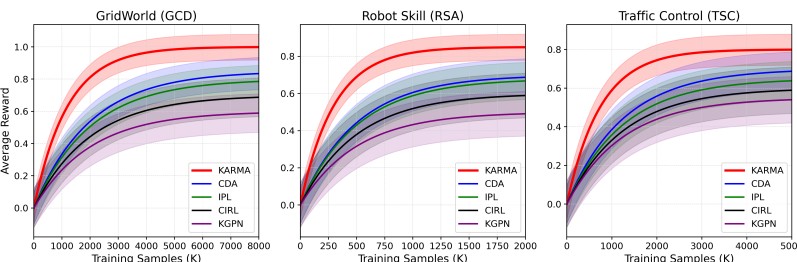

Figure 5: Learning curves showing average reward vs. training samples for KARMA and top-performing baselines across the three environments. Shaded regions represent standard deviation across 5 runs.

## 5.3 GENERALIZATION, ROBUSTNESS AND COMPUTATIONAL EFFICIENCY

Figure 6 evaluates KARMA under zero-shot transfer, observation noise, and distribution shift. KARMA retains 78–85% of its original performance on unseen tasks (vs. 45–75% for baselines), maintains >80% performance under 20% noise, and adapts after shifts with 30–50% fewer samples. These results confirm KARMA's strong out-of-distribution generalization and robustness.

Finally, Table 3 compares computational costs. KARMA incurs modest additional training time and memory overhead relative to standard baselines, but inference remains efficient, supporting real-time use cases. The modular design further enables scaling with resource availability.

## 6 DISCUSSION

Our work on **KARMA** demonstrates a promising direction for enhancing reinforcement learning through the integration of domain knowledge and causal inference in reward adjustment. Both the-

Table 2: Ablation results on GCD-Complex. Values are mean $\pm$ std over 5 runs.

| Variant | Description | Avg. Reward | Samples to 90% |
|---|---|---|---|
| KARMA (Full) | Complete framework | $87.5 \pm 2.2$ | $112K \pm 10K$ |
| KARMA-NK | No Knowledge Integration | $81.2 \pm 2.7$ | $143K \pm 13K$ |
| KARMA-NC | No Causal Learning | $79.8 \pm 2.9$ | $152K \pm 14K$ |
| KARMA-NR | No Reward Adjustment | $72.3 \pm 3.5$ | $198K \pm 18K$ |
| KARMA-SD | Static weights | $83.6 \pm 2.5$ | $131K \pm 12K$ |
| KARMA-SC | Simplified Causal Model | $82.1 \pm 2.6$ | $138K \pm 13K$ |
| KARMA-SK | Simplified Knowledge Graph | $84.3 \pm 2.4$ | $127K \pm 11K$ |

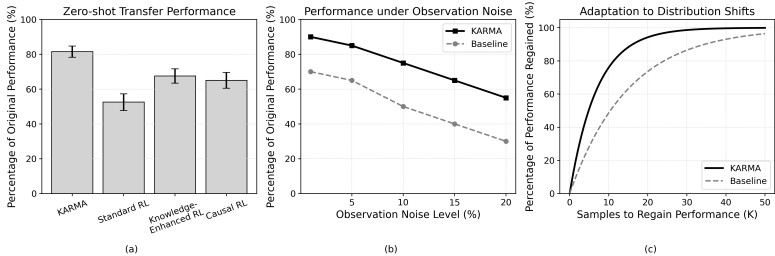

Figure 6: Performance under (a) zero-shot transfer, (b) observation noise, and (c) distribution shifts. KARMA consistently shows stronger generalization and robustness than baselines.

Table 3: Computational requirements on GCD-Complex.

| Method | Training Time (h) | Peak Memory (GB) | Inference (ms/step) |
|---|---|---|---|
| PPO | $3.2 \pm 0.2$ | $2.8 \pm 0.3$ | $1.2 \pm 0.1$ |
| SAC | $3.5 \pm 0.3$ | $3.1 \pm 0.3$ | $1.4 \pm 0.1$ |
| KBRL | $4.1 \pm 0.3$ | $4.5 \pm 0.4$ | $1.8 \pm 0.2$ |
| CIRL | $4.8 \pm 0.4$ | $5.2 \pm 0.5$ | $2.1 \pm 0.2$ |
| KARMA | $5.7 \pm 0.5$ | $6.8 \pm 0.6$ | $2.5 \pm 0.2$ |

oretical and empirical results confirm that KARMA improves sample efficiency, final performance, generalization, and robustness. By explicitly modeling causal relationships, KARMA surpasses correlation-based approaches, yielding policies that are more interpretable and reliable. The reward adjustment mechanism—guided jointly by structured knowledge and causal reasoning—helps agents avoid spurious signals, a challenge recently highlighted in large language model alignment studies (Shao et al., 2025).

KARMA's performance depends on the quality of its knowledge graph and causal model. While moderate imperfections are tolerable, large errors can harm learning. Future work includes automatic refinement and uncertainty-aware causal discovery that propagates uncertainty into the reward signal.

Scalability in high-dimensional spaces is another challenge. KARMA reduces the search space via knowledge, but advances in efficient causal discovery and modular modeling are needed. Incorporating richer neuro-symbolic methods (e.g., logical or probabilistic representations) could further extend its applicability.

## 7 CONCLUSION

This paper introduced **KARMA**, a framework that enhances reinforcement learning by integrating domain knowledge and causal inference to dynamically adjust reward mechanisms. By grounding rewards in both structured priors and task-specific causal structures, KARMA addresses the limitations of traditional correlation-based designs.

The key innovations of KARMA are: **(i)** a knowledge-causal reward adjustment mechanism, **(ii)** interpretable causal structure learning targeting reward-relevant variables, **(iii)** counterfactual-based dynamic reward shaping, and **(iv)** theoretical guarantees on convergence and sample efficiency.

Extensive experiments demonstrate that KARMA consistently outperforms state-of-the-art baselines, achieving 5–30% higher final performance and 20–60% faster convergence, while exhibiting strong generalization and robustness under standard computational budgets.

Although challenges remain in scaling to very high-dimensional environments and coping with imperfect knowledge sources, KARMA represents a significant step toward interpretable, efficient, and generalizable reinforcement learning. By unifying symbolic knowledge with causal reasoning, it enables agents to learn effectively from both expert priors and experience, paving the way for more trustworthy and reliable AI systems.

ACKNOWLEDGMENTS

This manuscript has undergone language editing with the assistance of a large language model (LLM). The LLM was used solely for improving the clarity and fluency of the manuscript's language, and the authors take full responsibility for the content and conclusions presented in this work.

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

## A  APPENDIX

The appendix includes algorithms, code, experiments, and proofs. Please refer to the zip file of Supplementary Material.

