# OpenReview forum: "KARMA: Knowledge-Aware Reward Mechanism Adjustment via Causal AI"
_ICLR.cc/2026/Conference — ICLR 2026 Conference Withdrawn Submission_

### Official Review · Reviewer_xMc8 · 2025-10-17

**Soundness:** 2
**Presentation:** 1
**Contribution:** 2
**Rating:** 2
**Confidence:** 3

**Summary:**

The paper proposes KARMA, a framework for reward adjustment in reinforcement learning that integrates domain knowledge and causal inference. The method aims to improve reward reliability and robustness by combining knowledge graphs, causal discovery, and counterfactual reasoning. Experiments are conducted on standard RL environments such as GridWorld, robotic control, and traffic signal control.

**Strengths:**

The paper addresses the long-standing issue of reward design and reliability in reinforcement learning—a topic that is becoming increasingly relevant in RLHF and safe RL research. The idea of grounding reward mechanisms in causal reasoning and knowledge priors is conceptually appealing and timely.

**Weaknesses:**

>1. Mismatch Between Motivation and Method Scope

The stated motivation of the paper centers on reward design issues in LLM reinforcement learning (e.g., RLHF/RLVR for LLMs). However, all proposed methods and experiments are conducted on small-scale, traditional RL tasks such as grid navigation and robot control.

This creates a disconnect between motivation and validation.
The framework is not applied or evaluated in large model settings, which weakens the paper’s claimed relevance to LLM alignment or reward modeling in high-dimensional generative environments.
The paper would be significantly strengthened if experiments or ablation studies were performed on an actual RLHF/RLVR setup, or at least through simulation that mimics language model reward alignment.

>2. Unclear Theoretical Analysis

The section titled “Theoretical Insights” provides only brief, informal claims about convergence and sample efficiency without derivations or proofs.
Even in appendix, I find the proof is not clear enough.

>3. Weak Experimental Comparison

The baselines are limited to traditional RL algorithms (PPO, SAC, TD3) and early causal/knowledge RL works (e.g., 2018–2021).
There is no comparison against modern large-model-based or hierarchical causal RL approaches, nor against any LLM-related RLHF/RLVR pipelines, despite that being part of the stated motivation.

>4. Writing and Presentation Issues

Abbreviations and acronyms are inconsistently defined and used.

Section fragmentation: excessive subheadings and “intuitive” remarks make the text disjointed and repetitive, reducing narrative flow.

Figures are visually small, and many lack detailed captions or axis labels. Figure quality needs improvement for clarity.

The writing includes several non-academic expressions and redundant descriptions, suggesting overreliance on automated language editing tools.

**Questions:**

Explain the weakness. And try to improve the quality from writing to experiments.

---

### Official Review · Reviewer_BhkX · 2025-10-30

**Soundness:** 2
**Presentation:** 2
**Contribution:** 2
**Rating:** 2
**Confidence:** 4

**Summary:**

This paper proposes KARMA, a causally informed reward adjustment framework. KAMRA modifies RL reward signals by extracting domain knowledge to form knowledge graph, constructing structural causal model, and counterfactual adjustment to improve training stability and efficiency.

**Strengths:**

1. The paper is well written.
2. The author conduct sufficient experiments to support their method.

**Weaknesses:**

1. Causal relationship extraction, structural causal model construction, and counterfactual reasoning have become the main research frameworks in causal AI. KARMA transfers this causal AI framework into RL rewards and makes targeted improvements to each part, but the improvements are incremental and marginal.
2. In Sec 3.1 and 3.2, using knowledge graph embedding to extract causal relationships is already a mature workflow. Therefore, KARMA's main innovation lies in using a mapping to search for the most relevant KG entities, but this point is essentially similar to attaching a Graph RAG to RL agent.
3. In Section 3.3, the main contribution of KARMA lies in the refinement of rewards through the addition of Knowledge-Based Reward and Causal Reward. However, more importantly, in the code of the supplementary materials, the authors use exponential decay and increase methods for weight transfer, which lacks strong theoretical or empirical experiments.
4. In Section 4.1, the authors set GCD 5 * 5 and 10 * 10 as the environment, but do not make a distinction report in Section 5.1.

**Questions:**

1. Figures should be presented in the form of vector graphics.
2. The readability of the article needs to be improved, and it is recommended that important content from many supplementary materials be reflected in the main text
3. In section 1.4, the author claims that KARMA can contribute to the generalization, safety, and alignment of modern RL methods. Can the author provide more detailed explanations, especially RL for LLMs?
4. I have checked the supplementary materials, and in fact, this paper should have a huge workload in the ablation experiment. In addition to the ablation of macro components, more factors should be considered, such as the search method for KG paths, the calculation of action embedding similarity, and the calculation of dynamic weights. Can the author provide a more comprehensive perspective?

---

### Official Review · Reviewer_sDBs · 2025-10-31

**Soundness:** 2
**Presentation:** 2
**Contribution:** 2
**Rating:** 2
**Confidence:** 3

**Summary:**

This paper proposes a framework called KARMA to address a core challenge in reinforcement learning (RL): reward design. The framework aims to address the problems of poor generalization and robustness caused by sparse, erroneous, or spurious reward signals. KARMA aims to improve the efficiency, robustness, and generalization of RL policies by introducing causal-aware reward shaping and counterfactual dynamic adjustment mechanisms. Experimental results demonstrate that KARMA achieves significant performance improvements and faster convergence compared to baseline methods in tasks such as grid navigation, robotic skill acquisition, and traffic control.

**Strengths:**

1. This paper proposes a unified framework that integrates domain knowledge, causal discovery, and dynamic reward adjustment, which is a very attractive and important conceptual vision.
2. The authors conducted experiments in three environments of varying complexity and compared KARMA with various baseline methods, including standard RL, knowledge-augmented RL, and causal RL. The results show the superiority of KARMA.

**Weaknesses:**

1. The overall structure of the paper is clear, and the high-level ideas and motivations are well conveyed to the reader. However, the paper is lacking in depth in key technical details, which affects the reproducibility and credibility of the work.
2. This framework integrates multiple computationally intensive components, such as online causal discovery and counterfactual querying. As can be seen in Table 3, its computational overhead is higher than all baselines. In environments with more complex, high-dimensional state-action spaces than those used in the experiments, the computational feasibility of this approach is a significant challenge.
3. Due to the lack of technical details and the direct application of existing technologies (KG, causal discovery algorithm), its algorithmic novelty is limited. Its main contribution is more reflected in the "integration" of the framework rather than "invention".
4. The entire framework relies heavily on building a knowledge graph and discovering causal graphs from online data. The paper does not discuss how the quality of the knowledge graph itself affects the framework.

**Questions:**

1. This paper's description of counterfactual reward adjustment ($R_{causal}$) is brief, with only mention of the use of "Pearl's do-calculus." Could you elaborate on this?
2. The description of the dynamic weight mechanism is insufficiently detailed; the paper mentions it as simply "linear annealing." This simple heuristic seems ill-suited to the complex problem it aims to solve. Why is linear annealing appropriate? How are its parameters set?

---

### Official Review · Reviewer_HRYb · 2025-11-01

**Soundness:** 1
**Presentation:** 1
**Contribution:** 2
**Rating:** 2
**Confidence:** 5

**Summary:**

The paper introduces KARMA, a causally-informed reward adjustment framework for reinforcement learning (RL). It aims to address the problem of spurious or poorly designed reward functions, which can lead to unstable learning and poor generalization. KARMA integrates structured domain knowledge to dynamically refine reward signals.

**Strengths:**

The paper makes an effort to apply causality tools in reinforcement learning problems. The proposed pipeline of integrating knowledge graphs and causal discovery algorithms with RL algorithms demonstrates interesting possibilities.

**Weaknesses:**

Overall, the paper's poor presentation renders it hard to evaluate the work's actual contribution, the proposed method's correctness, and the experiment results' significance. For the part I can evaluate based on sufficient information, I will expand into details as follows.
1. Factually wrong statements due to a lack of literature grounding. Firstly, it has been shown that the observation that random rewards could improve RLVR performances is due to dataset leakage [link](https://arxiv.org/html/2507.10532v1). Secondly, Causal RL, as also mentioned by the authors in the related work section, is a rapidly growing area, but many of the important works after 2021 are not cited or discussed. To name a few, [Sequential Causal Imitation Learning with Unobserved Confounders](https://openreview.net/forum?id=o6-k168bBD8), [Online Reinforcement Learning for Mixed Policy Scopes](https://proceedings.neurips.cc/paper_files/paper/2022/hash/15349e1c554406b7719d047a498e7117-Abstract-Conference.html), [Causal Dynamics Learning for Task-Independent State Abstraction](https://proceedings.mlr.press/v162/wang22ae/wang22ae.pdf). Noteably, the authors claim that "causal models...their systematic application for adjusting the reward mechanisms itself...remains underexplored". However, the following work in this year's ICML has explored how one can utilize causality tools to extract robust reward shaping functions from confounded offline data and should be discussed: [Automatic Reward Shaping from Confounded Offline Data](https://proceedings.mlr.press/v267/li25dr.html).

2. Those flow charts in the paper have a low information-to-noise ratio, while the main algorithm/theorems remain unclear. Will explain this point further in the questions section.

3. The proposed method doesn't set a clear boundary on what needs to be provided in the form of knowledge graphs, and to what extent one could expect meaningful performance gain without resorting to more exhaustive domain knowledge. The prior knowledge can be physical rules that dominate the robot's movements, can be grid world transition rules, or even optimal paths of a maze (this is actually used in this work's experiments). Either the answer is already trivially contained in the prior knowledge, or one can simply solve the problem by planning without any RL needed at all.

4. The experiment comparison is not fair due to the unrestricted knowledge graph prior. To put it simply, the current form of experiments is demonstrating that prior knowledge is useful in those state-based small environments with clearly defined object sets. I wonder if the authors also provide the knowledge graph augmented state observations to baselines like PPO, will they also do better? And can the proposed method still outperform that PPO variant, by how much?

5. Augmenting state space with knowledge graphs inherently limits the method's applicability to real-world problems. When considering language model training, the number of named entities in our daily language is simply too large to build a graph. Or for robotic tasks, there are tens of thousands of daily objects; it's not realistic that we label them all and define task-relevant attributes for each of them and for each new task.

**Questions:**

1. Could you provide more details on how the knowledge graph is constructed, with clear examples like those environments in your experiments? What are (h, r, t) (section 3.1)?

2. Fig.2 has multiple duplicated labels/nodes. Could you clarify?

3. Why do you need to do causal discovery in an MDP? Isn't MDP's causal diagram already fixed as S-->A, S,A-->Y , S,A-->S'? Are you assuming factored MDPs? Or more general SCMs? Or are you trying to learn the input factors to the causal reward from state + knowledge graph?

4. Sections 3.3 and 3.4 need serious rework. Could you provide more details on the algorithms you use to construct rewards and how those theories are derived intuitively?

5. I checked the appendix for proofs. For the sample efficiency part, what specific "standard RL algorithm" with such PAC guarantees do you assume when deriving the proof?  As far as I know, RL algorithms with PAC guarantees are not simple (like Delayed-Q, Q-UCB), and PPO is definitely not one of them. See [On the Theory of Policy Gradient Methods: Optimality, Approximation, and Distribution Shift](https://jmlr.org/papers/volume22/19-736/19-736.pdf).

---

### Note · Authors · 2025-11-19

I have read and agree with the venue's withdrawal policy on behalf of myself and my co-authors.